# Investigating trial design variability in trials of disease-modifying therapies in Parkinson's disease: a scoping review protocol

Marie-Louise Zeissler ![ORCID],[1,2] Timothy Boey,[3] Danny Chapman,[2] Gary Rafaloff,[4] Thea Dominey,[2] Karen G Raphael ![ORCID],[5,6] Susan Buff,[7] Hari Venkatesh Pai,[8] Emma King,[9] Paul Sharpe,[2] Fintan O'Brien,[2] Camille B Carroll ![ORCID] [1,2]

For numbered affiliations see end of article.

**Correspondence to**
Dr Marie-Louise Zeissler;
marie-louise.zeissler@plymouth.ac.uk

## ABSTRACT

**Introduction** Parkinson's disease (PD) is a debilitating neurological disorder for which the identification of disease-modifying interventions represents a major unmet need. Diverse trial designs have attempted to mitigate challenges of population heterogeneity, efficacious symptomatic therapy and lack of outcome measures that are objective and sensitive to change in a disease modification setting. It is not clear whether consensus is emerging regarding trial design choices. Here, we report the protocol of a scoping review that will provide a contemporary update on trial design variability for disease-modifying interventions in PD.

**Methods and analysis** The Population, Intervention, Comparator, Outcome and Study design (PICOS) framework will be used to structure the review, inform study selection and analysis. The databases MEDLINE, Web of Science, Cochrane and the trial registry ClinicalTrials.gov will be systematically searched to identify published studies and registry entries in English. Two independent reviewers will screen study titles, abstracts and full text for eligibility, with disagreements being resolved through discussion or by a third reviewer where necessary. Data on general study information, eligibility criteria, outcome measures, trial design, retention and statistically significant findings will be extracted into a standardised form. Extracted data will be presented in a descriptive analysis. We will report our findings using the Preferred Reporting Items for Systematic Reviews and Meta-Analyses Scoping Review extension.

**Ethics and dissemination** This work will provide an overview of variation and emerging trends in trial design choices for disease-modifying trials of PD. Due to the nature of this study, there are no ethical or safety considerations. We plan to publish our findings in a peer-reviewed journal.

## INTRODUCTION

Parkinson's disease (PD) is a progressive neurological disorder leading to debilitating motor and non-motor symptoms for patients.[1] It is the fastest growing neurological condition worldwide with cases projected to double by 2040.[2]

## STRENGTHS AND LIMITATIONS OF THIS STUDY

⇒ A key strength of this work will be its comprehensive nature ensured through the search validation process outlined in this publication.
⇒ The inclusion of English studies only could bias conclusions drawn from this work representing a limitation to this work.
⇒ Another limitation is the lack of universally adopted definitions for disease modification, which represents a risk for misclassification of trials within this review.
⇒ To mitigate this, we have developed clear guidance for classification of trials via a decision tree and will adopt a consensus review process for study screening.
⇒ Deep brain stimulation studies will be excluded from the review.

Although many symptoms can initially be treated effectively by dopamine replacement therapies,[3] no disease-modifying therapies (DMTs) have been identified to slow, stop or reverse progression of PD since the first DMT trial for selegiline in 1989.[4] It is possible that negative late phase studies reflect a genuine ineffectiveness of treatments, stemming from the lack of translatability of preclinical models to the clinic. However, phase 2 trials have demonstrated signals of efficacy which were then not translated into positive results at phase 3.[5–8] Thus, failure at both phases 2 and 3 could be a consequence of trial methodology leading to false-positive or false-negative results including parameters such as small sample size or inadequately compensating for known challenges of DMT trial design in PD such as the lack of biomarkers that correlate with clinical disease progression,[9] the heterogeneity of the disease course,[10–12] placebo effects and symptomatic therapy complicating the measurement of disease progression.[13]

The development of an effective design for the testing of DMTs is critical and has been the subject of ongoing debate leading to a number of recommendations for more effective trial designs. These include more refined eligibility criteria targeting more homogeneous patient populations (such as early PD or genetic subtypes), longer trial durations and outcome measure alternatives to the Movement Disorder Society-Unified Parkinson's Disease Rating Scale (MDS-UPDRS).[13 14] However, it is unclear to what extent such methods have been adopted within the last 33 years and whether there are indications of some trial design strategies being more effective than others.

Two previous systematic reviews by Hart *et al* in 2009 and McGhee *et al* in 2016 as well as recent reports by McFarthing *et al* show that there is a rich landscape of DMT trials in PD[4 15–17] providing a potentially rich dataset to chart different trial designs.

Improved understanding of the pathogenesis of PD, combined with advances in silico approaches, have led to an accelerated rate of drug discovery as well as targeted drug-repurposing programmes[18 19] resulting in an expansive clinical research pipeline for DMTs.[15] More efficient approaches to test new therapies are needed to allow for the increasing number of promising therapies to be investigated in a timely manner. One such approach is that of the adaptive multi-arm, multi-stage (MAMS) platform trial, which is currently being developed for PD through the Edmond J Safra Accelerating Clinical Trial in Parkinson's Disease (EJS ACT-PD)

initiative and aims to accelerate clinical testing of novel therapies.[20]

Here, we report on our protocol to systematically chart the design of phases 2 and 3 disease-modifying trials in PD with the view of informing the design of a randomised-controlled phase 3 adaptive MAMS platform trial for DMTs in PD. The review will provide an overview of trial design characteristics such as participant selection, stratification/minimisation criteria, trial size, duration and outcome measures to assess whether there are emerging trends on trial design choices.

## METHODS AND ANALYSIS

The scoping review protocol presented here was written in accordance with Preferred Reporting Items for Systematic Reviews and Meta-Analyses Scoping Review guidelines.[21] The Population, Intervention, Comparator, Outcome and Study design (PICOS) framework[22] will be used to structure the review, inform study selection and analysis.

Herein, we outline our planned approach for literature search, article selection, data extraction and charting.

### Inclusion criteria for study selection

We have used the PICOS design framework to develop study eligibility criteria aiding in the identification of PD trials (table 1). Records in English, including published and planned as well as unpublished studies identified

**Table 1** PICOS design framework

| PICOS domain | Eligibility criteria |
|---|---|
| Population | Participants with idiopathic PD |
| Intervention | Only studies investigating DMTs will be included. Studies whose sole purpose is the improvement of symptoms will be excluded. We will identify articles through one of the following two methods:<br>1. A stated intent of the authors to study a neuroprotective effect (such as through a rationale of prevention or restoration of pathology) or disease-modifying effect (such as an intent to delay disease progression or development of clinical milestones) within the publication or study registry entry. We will carefully consider titles, abstracts and introductions of publications to judge the author's intent as there are no ubiquitously used terminology conventions or Medical Subject Headings terms for DMTs within the field.<br>Studies with known symptomatic effects, such as selegiline, rasagiline and pramipexole will be included provided the primary intent of the authors is to evidence disease modification or neuroprotection within the study.<br>2. A literature search of the intervention revealing that the intervention has only been studied in the context of disease modification or neuroprotection.<br>Studies investigating deep brain stimulation will be excluded. |
| Comparator | Included studies will have to be randomised and controlled with comparators being clearly identified by the authors as a control condition. Both open label and placebo-controlled trials will be included. No restrictions on types of control conditions will be imposed allowing for the inclusion of both open label and placebo-controlled trials. |
| Outcome | The focus of the review is on phases 2 and 3 efficacy trials and therefore trials will have to include at least one efficacy outcome. Pure safety trials will be excluded. |
| Study design | Only phases 2 and 3 trials will be included as this work will be carried out to support the design of a phase 2/3 platform trial. For article screening purposes, trial phases as stated by article or registry entry will be used. |

DMTs, disease-modifying therapies; PD, Parkinson's disease; PICOS, Population, Intervention, Comparator, Outcome, and Study.

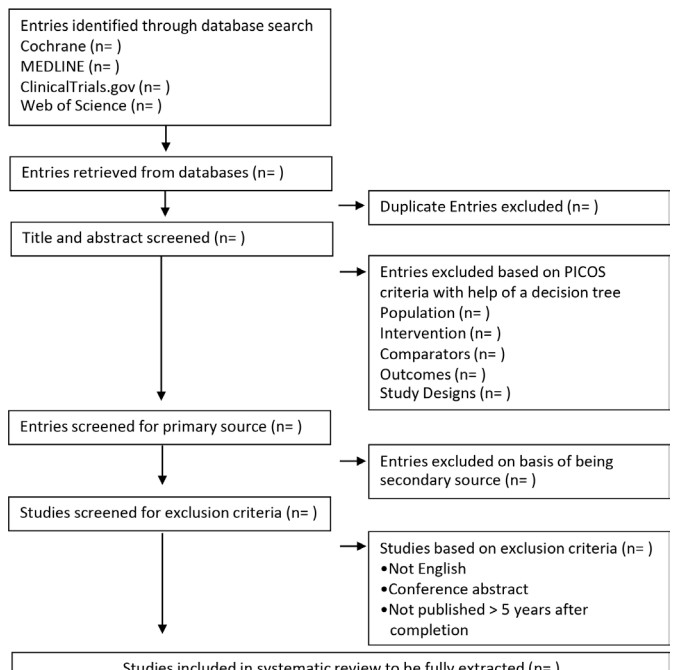

**Figure 1** Flow diagram outlining the selection procedure to identify randomised-controlled trials included within the study.

within ClinicalTrials.gov, will be fully extracted. Phase 1 studies will be excluded as the focus of the review is to inform the design of a phase 2/3 study seeking to evidence efficacy rather than safety/tolerability which is the focus of phase 1 studies. Studies for which only conference abstracts are available will be excluded as information within abstracts is too limited for data extraction. A flowchart of planned article selection is presented in figure 1.

### Search methods for identification of studies

Searches will be carried out in MEDLINE, Web of Science, Cochrane and ClinicalTrials.gov from inception to 1 October 2023 as outlined in online supplemental file 1.

Searches were developed using the following validation methodologies.

### Step 1: Identification of a random sample of published articles meeting study eligibility criteria

ClinicalTrials.gov was searched using the following fairly indiscriminate search parameters: Study status: Recruiting, Not yet recruiting, Active, not recruiting, Completed, Enrolling by invitation, Suspended, Terminated, Withdrawn and Unknown status Studies; Study type: Interventional Studies; Condition or disease: Parkinson Disease; and Phase: 2,3,4 and screened for articles meeting the outlined eligibility criteria (table 1).

To identify a random sample of published articles that would be eligible for study inclusion, Clinical-Trials.gov entries were screened using a decision tree (online supplemental file 2) based on PICOS criteria outlined in table 1. Published articles were sought for all eligible entries whose ClinicalTrials.gov status was marked as 'completed' as this subset of entries has the highest chance of having an associated published article.

### Step 2: Identification of keywords for searches

Search strategies for MEDLINE, Web of Science and Cochrane were built using keywords associated with published articles identified in step 1. In addition, common phrases used to describe disease modification trials were identified from published abstracts.

### Step 3: Search strategy optimisation

Using DOIs for relevant studies identified in step 1, we established how many of these published articles were present in each database. The effectiveness of search term combinations for each database was then evaluated by calculating the percentage of relevant DOIs found by each search iteration versus those known to be present within the database. We aimed for a search efficiency of higher than 70%.

Search terms identified through this validation process are presented in table 2. Full search strategies developed as above can be found in online supplemental file 1. Search strategies will be peer reviewed following PRESS guidelines.[23]

| Category | Keywords | Additional common words (in abstract or title) | Additional parameters based on most common non-relevant hits |
|---|---|---|---|
| Population | Parkinson's disease OR human OR patients OR aged | Subject* OR Participant* | |
| Intervention | Therapy OR Disease Progression | Neuroprotect* OR Delay* OR Improv* OR Treatment | |
| Comparator | Random allocation OR Control groups OR placebo | | |
| Outcome | Safety OR adverse | Efficacy OR benefit OR slow OR risk | NOT 'deep brain stimulation' NOT 'predict* model' |
| Study design | Clinical trial | Study OR Phase | |

**Table 2** Electronic search keywords

## Study selection

Searched studies will be screened by two independent reviewers blinded to each other's decisions. A screening decision tree will be used (online supplemental file 2) to standardise decision-making in line with the PICOS criteria outlined in table 1. The relevant decision tree step number will be recorded as reasoning for include/exclude decisions. Disagreements will be resolved through common consensus after a discussion. On sustained disagreement, a third expert reviewer opinion will be sought.

## Data extraction and management

General study information as well as three extraction domains (eligibility criteria, study outcome measures, study design) will be extracted from the main publications as well as information held on trial registries and recorded in a predetermined form featuring the fields outlined in table 3. It is anticipated that more than one source of information will exist for some studies (registry entry and publication). Referenced, raw text will be extracted alongside the final data field to facilitate data entry and amalgamation of conflicting data from different sources. The following hierarchy will be used for handling data source contradictions: peer-reviewed primary results paper will be classed as the most trustworthy source, followed by peer-reviewed secondary results papers, then protocol papers and finally registry entries. Data for each section will be extracted by one reviewer. An independent reviewer will cross-check ≥20% of the extracted data for each extraction domain. Where extracted data differs between reviewers, discussions to form a common consensus will be held. Prominent levels of discrepancy will be reviewed and may lead to a greater extent of double extraction, better definition of data extraction fields or the consultation of a third expert reviewer. Non-reported data will be recorded as 'Not Specified'. Raw data reported in the results paper will be made available as a supplement or within an appropriate data repository.

## Extracting and charting results

### Study phase

Where possible, we will separate reporting and analysis of phases 2 and 3 trials. We anticipate some reporting heterogeneity of phase classification due to poor definitions or overlapping interchangeable concepts. Phases stated as 1–2, 2–3, 2A, 2B and 2 classed will be classified as phase 2 trials and phases stated as 3 or 3–4 will be classed as phase 3 trials.

### Trial success

Trial success will be recorded as studies showing a statistically significant result for a primary outcome. It is likely that, especially in phase 2 studies and studies with no corresponding registry entry, primary outcomes may not always be stated clearly; where this is the case, all outcomes will be treated as co-primary outcomes. Where only one

**Table 3** Data to be extracted

| Extraction domain | Data to extract |
|---|---|
| General study information | Intervention studied* |
| | Status of study* |
| | Year of publication* |
| | Year of registration† |
| | Year of completion/termination* |
| | Named sites† |
| | Number of countries† |
| | Lead site country† |
| Eligibility criteria | Age limits† |
| | Disease duration* |
| | Hoehn and Yahr stage* |
| | Hoehn and Yahr on/off state* |
| | Inclusion criteria present: cognition† |
| | Definition of cognition criterion† |
| | Inclusion criteria present: depression† |
| | Definition of depression score† |
| | Inclusion criteria present: drug naive* |
| | PD drug stability† |
| | Changes to PD drugs permitted?† |
| Outcome measures | Primary outcome measures* |
| | Other outcome measures* |
| | Outcome domains* |
| Study design | Primary endpoints met* |
| | Other endpoints met* |
| | Phase of trial* |
| | Number of sites* |
| | Number of arms* |
| | Number of participants enrolled/estimated* |
| | Attrition (control arm)* |
| | Attrition (active arm)* |
| | Level of blinding† |
| | Type of control* |
| | Stratification parameters† |
| | Wash out present† |
| | Wash in present† |
| | Overarching design type and details† |
| | Dose ranging† |
| | Study duration (baseline to final visit)* |
| | Number of follow-ups* |
| | Follow-up frequency* |
| | Treatment extension† |

*Required for planned analyses.
†Other exploratory extraction fields.
PD, Parkinson's disease.

of many co-primary outcomes shows a statistically significant result, partial success will be recorded.

For this analysis, each independent study/trial will be considered one unit of analysis.

### Eligibility criteria

We will report on the proportion of studies investigating interventions in early versus late PD populations. For this purpose, we will define an early PD population as studies specifying study eligibility as people with PD with disease duration≤5 years or Hoehn and Yahr stage≤2.5 or participants being drug naive (diagnosed but not yet having received any dopamine replacement medications for their PD) as criteria for study inclusion. We defined these cut-offs based on the interrogation of data from a preliminary literature review conducted by us[24] as being commonly used by researchers to self-identify studies as targeting an 'early PD' population. Furthermore, impairment of postural reflexes marked by the reaching of Hoehn and Yahr stage 3 has been linked to disability and is of meaningful impact to patients in terms of quality of life,[25] thereby defining a distinct later stage of PD.

### Outcome measures

All outcome measures will be extracted. We will distinguish, where possible, between primary outcome measures and other outcome measures. There will be no further classification into secondary or exploratory outcome measures as this is likely to be inconsistently reported in both registry entries and published articles. We will provide full data on frequency of all outcome measures and will summarise these as follows: the frequency of outcome domains used as primary outcome measures in phases 2 and 3 trials. Outcome domains will be defined using the National Institute for Neurological Disorders and Stroke Common Data Elements (NINDS-CDE) for PD domains and subdomains.[26] We will additionally chart the use of the most common outcome measure scale, UPDRS and the MDS-UPDRS, reporting on the use of its parts and part combinations as primary outcome. This is particularly important in the light of a recent report by the scale authors' affirming a recommendation against the combination of part 3 with other parts of the scale.[27]

### Outcome measures success

We will perform a descriptive analysis to summarise the variety of primary outcome measures used and the proportion that reached statistical significance. Depending on the variety of outcome measures found through the review, outcome measures may be grouped according to NINDS-CDE outcome domains or subdomains. Outcome measures found to be statistically significant will be recorded for all completed and reported studies. Here, each primary outcome measure reported in a study result publication will be considered as a unit of analysis. This will allow insights into whether and which primary outcome measures have been particularly successful in trials.

### Study size, duration, follow-ups and attrition

We will perform a descriptive analysis to summarise study size, duration, number and frequency of follow-ups and attrition. This will allow insights into the impact of study size, length and assessment burden on retention within DMT trials.

### Study design trends over time

Study design characteristics will be analysed for overall frequency of occurrence and changes in frequency over time with each independent study/trial being considered one unit of analysis.

### Assessment of reporting biases

We will report on the number of studies that have been completed for longer than 5 years without a published peer-reviewed results report to provide an indication of potential reporting bias.

### Patient and public involvement

Two patients (GR and KR) and one carer (SB) were involved in the design and conduct of the study and are coauthors of this manuscript. Additionally, they have an impact on the scope of the work by advocating for inclusion of non-pharmacological interventions within the review.

## ETHICS AND DISSEMINATION

Due to the nature of this study, there are no ethical or safety considerations. The full results of this study will be published in a peer-reviewed journal. Extracted data relevant to the published analysis will be made available as a supplement to the main results publication, alongside data sources such as registry entries and publication DOIs or deposited in an appropriate data repository.

## DISCUSSION

Our systematic review of DMT trial design for PD aims to explore the variation of trial design choices and where consensus might be emerging for phase 2 and phase 3 study design. Currently, no DMTs have passed the hurdle of phase 3 success and therefore there has been no update to standard of care for PD beyond refinement of symptomatic therapy options.

By employing a search validation methodology and aiming for a search efficiency of higher than 70% in all databases, this review will produce a comprehensive overview of past DMT trials, on which our assessment of emerging trends in PD DMT trial design is based.

The restriction to data sources written in English language represents a limitation of this study.

Our review will provide a comprehensive overview of potential design choices to consider for future trials, including the EJS ACT-PD MAMS platform trial.[28]

### Author affiliations
1Newcastle University, Newcastle upon Tyne, UK

[2]Faculty of Health, University of Plymouth, Plymouth, UK
[3]School of Medicine, University of Liverpool, Liverpool, UK
[4]Parkinson's Research Advocate, Westlake, Florida, USA
[5]Oral & Maxillofacial, Radiology and Medicine, New York University, Brooklyn, New York, USA
[6]Parkinson's Research Advocate, New York, New York, USA
[7]Parkinson's Research Advocate, Sunnyvale, California, USA
[8]Queen's Medical Centre, Nottingham, UK
[9]University Hospitals Plymouth NHS Trust, Plymouth, UK

**Correction notice** This article has been corrected since it was published. Licence changed to CC BY on 15/01/24.

**Contributors** M-LZ and TB: manuscript drafting; M-LZ, TB, DC, GR, HVP, PS, EK and FO'B: validation of search methods, development and trial of decision tree tool and data table construction; SB, GR, KR, TD, CBC and M-LZ: study design; CBC and M-LZ: study oversight; and all authors: input into manuscript.

**Funding** This study was funded by Cure Parkinson's Grant number CC021. The funder was not actively involved in the development of this protocol.

**Competing interests** None declared.

**Patient and public involvement** Patients and/or the public were involved in the design, or conduct, or reporting, or dissemination plans of this research. Refer to the Methods section for further details.

**Patient consent for publication** Not applicable.

**Provenance and peer review** Not commissioned; externally peer reviewed.

**ORCID iDs**
Marie-Louise Zeissler http://orcid.org/0000-0002-6232-4284
Karen G Raphael http://orcid.org/0000-0002-2804-8124
Camille B Carroll http://orcid.org/0000-0001-7472-953X

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
