## [Reviewer comments · BMJ Open]

ARTICLE DETAILS

TITLE (PROVISIONAL)	Investigating trial design variability in trials of disease-modifying therapies in Parkinson's disease: a scoping review protocol
AUTHORS	Zeissler, Marie-Louise; Boey, Timothy; Chapman, Danny; Rafaloff, Gary; Dominey, Thea; Raphael, Karen; Buff, Susan; Pai, Hari Venkatesh; King, Emma; Sharpe, Paul; O' Brien, Fintan; Carroll, Camille

VERSION 1 – REVIEW

REVIEWER	Sampaio, Cristina CHDI Foundation
REVIEW RETURNED	11-Apr-2023

GENERAL COMMENTS	The manuscript describes the protocol for a planned systematic review aiming to investigate trial design variability in disease-modifying therapies for Parkinson's disease. The primary focus is on trial design characteristics rather than the results of the trials, as clearly stated in the title and throughout the paper. However, the abstract needs revision to avoid ambiguity about the primary goal of the systematic review. In the introduction, it is suggested that positive results of phase 2 trials are always true positives, but it is well known that due to their smaller sample size, phase 2 trials are prone to false positives. Therefore, the statement needs to be revised to reflect this fact accurately. Regarding the inclusion criteria, the population must have clinically diagnosed PD, but it is unclear which diagnostic criteria will be accepted. Additionally, the terms "disease-modifying" and "neuroprotective" are used interchangeably, despite the theoretical possibility of therapies that rescue dysfunction rather than preventing it. The inclusion criteria should also clarify that therapies with known symptomatic effects, such as selegiline, rasagiline, and pramipexole, will still be considered. The exclusion criteria exclude deep brain stimulation, but it is unclear whether other interventions, such as lesion surgery, Focal Ultrasound Thermal Lesions (FUS), or transplantation, will be excluded. In the data extraction section, it is important to state how full paper publication and protocols provided as supplementary material will be handled. Contradictions between the published paper and the protocol are not uncommon.
---

	Additionally, in Table 3, the eligibility criteria are listed, but it is important to compare these criteria with the characteristics of the population actually recruited, as sometimes the latter has substantial deviations from eligibility criteria. For example, the eligibility criteria may allow recruitment up to the age of 80, but the actual recruited population may not exceed 62 years old.
--	---

REVIEWER	Hirschwald, Julia Trinity College Dublin, Clinical Speech and Language Studies
REVIEW RETURNED	01-May-2023

GENERAL COMMENTS	Thank you very much for the opportunity to review this manuscript for a systematic review protocol on trial design variability in trials of disease-modifying therapies in Parkinson's disease. The protocol is well designed and written. I have the following comments to strengthen the quality of the protocol and ultimately the review: Abstract  Line 10 "sensitive, objective outcome measures" – I would suggest writing "validated outcome measures" instead Lines 11-13: the objective stated in the abstract does not match the objective stated at the end of the introduction; I suggest amending this to make sure these are correct and identical Lines 15-17: The reporting guidelines are PRISMA-P; therefore, I would suggest writing "We will be analysing our findings according to PICOS and report them following PRISMA-P" Line 19: clinicaltrials.gov is not a database, this is only a trial registry; I suggest amending this throughout the manuscript Strengths and Limitations One limitation of the review will be that you only search for studies written in English. This is a potential bias; therefore, I suggest listing this as a limitation. Introduction  Line 23: It would be beneficial to provide a definition of disease-modifying therapies so that it is clear what you refer to. Line 26: You have introduced the abbreviation for Parkinson's disease (PD) in line 16. I suggest only using the abbreviation PD from now on to be consistent throughout the manuscript. Line 30: It is unclear why 32 years and from what year on; I suggest clarifying this. Lines 54-60: As the reader it would be interesting to learn more about the platform and its aims. Maybe you could explain this in more detail here and provide a stronger rationale for the need of this review. Methods and analysis Based on the aims of this review, a scoping review seems more suitable. I suggest considering conducting a scoping review or provide a strong rationale for conducting a
---

	systematic review. The article by Munn et al. (2018) might be helpful https://doi.org/10.1186/s12874-018-0611-x  Lines 8-9: As suggested in the abstract, I would write “The PICOS framework will be used to analyse the findings” Lines 10-11: This sentence does not fit here, so I suggest omitting it. Line 14: Figure 1 belongs to “selection process” (PRISMA-P #11), this can be omitted here. Inclusion criteria for study selection  Line 21: You state that you will include unpublished PD trials. However, you do not include grey literature. Could you please clarify how you will search for unpublished PD trials? Line 22: “unpublished for a maximum of 5 years following study completion” – This is unclear, I suggest explaining this and giving a rationale. Line 23: Why are phase 1 studies and conference abstracts excluded? Could you provide a justification for this? Table 1 population: I suggest shortening the inclusion criterion to “participants with idiopathic PD” Table 1 intervention: Please provide a rationale for excluding DBS Table 1 comparator, outcome, study design: It would be helpful to provide a definition for phase 2 and 3 trials as these are used differently in the literature. Regarding the outcomes: please also define efficacy and safety outcomes and provide a rationale for excluding pure safety trials as phase 2 trials are safety trials by nature. Search Methods for Identification of Studies  Line 17: please update the date and search as this is a protocol this should be dated after publishing the protocol Lines 20-25: This is a protocol; therefore, the search should not have been conducted yet. Furthermore, please use future tense for what you are planning to do. Lines 24-26: I suggest writing “inclusion criteria” instead of “PICOS criteria” Line 29: You write that only studies with a status marked as “completed” will be included. Does this mean that you will exclude studies where the status was possibly not updated to “completed” even though the study has been completed and possibly the results are already published? It would be helpful if you could clarify this. Line 38: Is this validation process based on the PRESS (peer review of electronic search strategies) guidelines? If so, I would suggest referencing these guidelines and explaining in more detail how you will implement this. Data extraction and management
--	--

	1. Table 3 outcome measures: It is unclear what you mean by “outcome domain”. Do you have a reference for this? E.g. Dodd et al. 2018? It would be helpful to clarify this. 2. Table 3 outcome measures: Are you only looking at outcome measures or at outcomes or at both? It might be that you mean outcomes instead of outcome measures? 3. Table 3 study design: It could be interesting to look at the frequency of measurement in addition to study duration; so how many follow-ups did they conduct and at what timepoints. Risk of bias assessment 1. The Jadad scale seems to be oversimplistic as there is too much emphasis on blinding. I suggest choosing another tool or provide a strong rationale for using the Jadad scale. 2. Line 56: Why are you planning on analysing only selected studies for risk of bias? Please provide a strong rationale for this and explain how you will select these. Data collection and planned analysis 1. Lines 15-19: This is another aim / objective. I suggest moving this to the introduction section and align it with the objective in the abstract. 2. Eligibility criteria, lines 45-47: Please provide the reference for this scoping review 3. Outcome measures lines 54-55: In table 3 you differentiate between primary and secondary outcomes; therefore, this is contradictory. I suggest providing a clarification or amending this section accordingly. 4. Outcome measures line 59: Please explain how you will analyse this. Additionally, in the introduction you stated that there might be more suitable, validated scales other than the UPDRS. What is your rationale for only analysing the two listed scales? Development of consensus It is unclear how the stated description relates to the development of consensus. I suggest amending this paragraph. Assessment of Reporting Bias Please provide a rationale for excluding these trial registry entries. I recommend considering that if you do exclude them, this could add to a risk of reporting bias in your review. Patient and Public Involvement 1. Line 37: I assume that you mean two instead of three patients? Please adapt accordingly. Funding statement According to the PRISMA-P checklist #5c please describe the role of funder(s), sponsor(s) and/or institution(s), if any, in developing the protocol. Acknowledgements for this peer review
--	---

	I would like to acknowledge Prof. Margaret Walshe (Associate Professor, Clinical Speech and Language Studies, Trinity College Dublin, Ireland), who supported me in completing this peer review
--	---

VERSION 1 – AUTHOR RESPONSE

Reviewer 1

Number	Comment	Response
1	The primary focus is on trial design characteristics rather than the results of the trials, as clearly stated in the title and throughout the paper. However, the abstract needs revision to avoid ambiguity about the primary goal of the systematic review.	The abstract was amended to clarify the review's focus on trial design variability for disease-modifying interventions in PD. (page 2 para 1 and 3)
2	In the introduction , it is suggested that positive results of phase 2 trials are always true positives, but it is well known that due to their smaller sample size, phase 2 trials are prone to false positives. Therefore, the statement needs to be revised to reflect this fact accurately.	The statement was revised to: "Thus, failure at both phase 2 and 3 could be a consequence of trial methodology leading to false positive or negative results including parameters such as small sample size or inadequately compensating for known challenges of DMT trial design in PD such as the lack of biomarkers that correlate with clinical disease progression (9), the heterogeneity of the disease course (10-12), placebo effects and symptomatic therapy complicating the measurement of disease progression (13). " (page 3, para 7)
3	Regarding the inclusion criteria , the population must have clinically diagnosed PD, but it is unclear which diagnostic criteria will be accepted.	There will be no restriction based on the specific diagnostic criteria used by investigators as these have changed within the last 30 years and there may be international differences. We have adjusted the wording of table 1 as requested by reviewer 2 to: "Participants with idiopathic PD" (page 5 table 1)
4	Additionally, the terms "disease-modifying" and "neuroprotective" are used interchangeably, despite the theoretical possibility of therapies that rescue dysfunction rather than preventing it.	Thank you for this comment. Our preliminary searches revealed that this terminology is also interchangeably used within trial descriptions by investigators. We have further defined our definition to: "Only studies investigating disease modifying therapies will be included. Studies whose sole purpose is the improvement of symptoms will be excluded. We will identify articles through one

		of two methods (page 5, table 1 (intervention)): 1) A stated intent of the authors to study a neuroprotective effect (such as through a rationale of prevention or restoration of pathology) or disease modifying effect (such as an intent to delay disease progression or development of clinical milestones) within the publication or study registry entry. We will carefully consider titles, abstracts and introductions of publications to judge the author's intent as there are no ubiquitously used terminology conventions or MeSH terms for DMTs within the field.” We acknowledge that this definition still leaves a small possibility of uncertainty around the classification of interventions within the review and we have added this as a listed limitation (page 3).
5	The inclusion criteria should also clarify that therapies with known symptomatic effects, such as selegiline, rasagiline, and pramipexole, will still be considered.	We included the following statement within table 1, Intervention (page 5): “ Studies with known symptomatic effects, such as selegiline, rasagiline, and pramipexole will be included provided the primary intent of the authors is to evidence disease modification or neuroprotection within the study.”
6	The exclusion criteria exclude deep brain stimulation, but it is unclear whether other interventions, such as lesion surgery, Focal Ultrasound Thermal Lesions (FUS), or transplantation, will be excluded.	Thank you for this comment. Our preliminary searches identified deep brain stimulation as a frequent non-relevant hit. Whilst it is possible that the introduction of this exclusion may introduce a small chance of missing DBS related disease modification studies, this decision was mainly based on practicality. We have added this to the study limitations (page 3). All other interventions will be judged on authors' intent. We have not made any further changes to the manuscript as this is outlined in table 2 (page 7) which describes the rationale for electronic search keywords.
7	In the data extraction section, it is important to state how full paper publication and protocols provided as supplementary material will be handled. Contradictions between the published paper and the protocol are not uncommon.	We have clarified our description for the handling of data source discrepancies: “The following hierarchy will be used for handling data source contradictions: Peer reviewed primary results paper will be classed as the most trustworthy source, followed by peer reviewed secondary results papers, then protocol papers, and finally registry entries. “

		page 7, para 2
8	Additionally, in Table 3, the eligibility criteria are listed, but it is important to compare these criteria with the characteristics of the population actually recruited, as sometimes the latter has substantial deviations from eligibility criteria. For example, the eligibility criteria may allow recruitment up to the age of 80, but the actual recruited population may not exceed 62 years old.	Many thanks for this comment we have decided to change this review to a scoping review with the primary focus on charting trial design choices made by researchers and thus the extraction of recruited population characteristics will be outside of the scope of this review. See reviewer 2, comment 10.

Reviewer 2

	Comment	Response
Number	Abstract	
1	Line 10 “sensitive, objective outcome measures” – I would suggest writing “validated outcome measures” instead	We thank the reviewer for this suggestion, however we would like to retain these two words as in Parkinson’s the main issue is not the lack of validated outcomes but their sensitivity to change in a disease modification setting, where expected effects are much more modest. This has been reworded slightly to clarify (page 2, para 1): “Diverse trial designs have attempted to mitigate challenges of population heterogeneity, efficacious symptomatic therapy and lack of outcome measures that are objective and sensitive to change in a disease modification setting”
2	Lines 11-13: the objective stated in the abstract does not match the objective stated at the end of the introduction; I suggest amending this to make sure these are correct and identical	We have amended this as suggested (page 2, para 1). “Here we report the protocol of a scoping review that will provide a contemporary update on trial design variability for disease-modifying interventions in PD.”
3	Lines 15-17: The reporting guidelines are PRISMA-P; therefore, I would suggest writing “We will be analysing our findings according to PICOS and report them following PRISMA-P”	We have changed the review to a scoping review and have amended the abstract accordingly. (page 2, para 2 and 3): “The Population, Intervention, Comparator, Outcome and study design framework (PICOS) will be used to structure the review, inform study selection and analysis.”

		“ We will report our findings using the Preferred Reporting Items for Systematic Reviews and Meta-Analyses (PRISMA) Scoping review extension.”
4	Line 19: clinicaltrials.gov is not a database, this is only a trial registry; I suggest amending this throughout the manuscript	Any mention of clinicaltrials.gov as a database was changed to registry or the distinction removed as appropriate in the abstract and throughout the manuscript.
	Strength and Limitations	
5	One limitation of the review will be that you only search for studies written in English. This is a potential bias; therefore, I suggest listing this as a limitation.	This was added to the strengths and limitations section as well as the discussion (Page 3): “The inclusion of English studies only could bias conclusions drawn from this work representing a limitation to this work.”
	Introduction	
6	Line 23: It would be beneficial to provide a definition of disease-modifying therapies so that it is clear what you refer to.	We have reworded this line to (page 3): “Although many symptoms can initially be treated effectively by dopamine replacement therapies (3), no disease modifying therapies (DMTs) have been identified to slow, stop or reverse progression of Parkinson’s since the first DMT trial for selegiline in 1989 (4). “
7	2. Line 26: You have introduced the abbreviation for Parkinson’s disease (PD) in line 16. I suggest only using the abbreviation PD from now on to be consistent throughout the manuscript.	Reference to Parkinson’s or Parkinson’s disease was changed to PD throughout the manuscript.
8	3. Line 30: It is unclear why 32 years and from what year on; I suggest clarifying this.	This was clarified by changing the sentence to (page 3): “Although many symptoms can initially be treated effectively by dopamine replacement therapies (3), no disease modifying therapies (DMTs) have been identified to slow, stop or reverse progression of Parkinson’s since the first DMT trial for selegiline in 1989 (4)”
9	4. Lines 54-60: As the reader it would be interesting to learn more about the platform and its aims. Maybe you could explain this in more detail here and provide a stronger rationale for the need of this review.	We have moved the paragraph on platform aims towards the end of the introduction to emphasise the aim of informing the design of the Multi-arm multi-stage (MAMS) platform trial (page 3-4)
	Methods and analysis	

10	Based on the aims of this review, a scoping review seems more suitable. I suggest considering conducting a scoping review or provide a strong rationale for conducting a systematic review. The article by Munn et al. (2018) might be helpful https://doi.org/10.1186/s12874-018-0611-x	We thank the reviewer for this insight we have taken into account the comments by both reviewers and concluded that the evidence synthesis proposed for this article may not be strong enough to warrant the production of a systematic review at this stage. We have revised the manuscript in accordance with scoping review reporting guidelines: Tricco, AC, Lillie, E, Zarin, W, O'Brien, KK, Colquhoun, H, Levac, D, Moher, D, Peters, MD, Horsley, T, Weeks, L, Hempel, S et al. PRISMA extension for scoping reviews (PRISMA-ScR): checklist and explanation. Ann Intern Med. 2018,169(7):467-473. doi:10.7326/M18-0850 .
11	1. Lines 8-9-: As suggested in the abstract, I would write “The PICOS framework will be used to analyse the findings”	We thank the reviewer for this suggestion. We have changed the wording to (page 4, para 5): “The Population, Intervention, Comparator, Outcome and study design framework (PICOS) will be used to structure the review (22), inform study selection and analysis”
12	2. Lines 10-11: This sentence does not fit here, so I suggest omitting it.	The sentence was deleted (page 4, para 5).
13	3. Line 14: Figure 1 belongs to “selection process” (PRISMA-P #11), this can be omitted here.	Mention of Figure 1 was moved to the section of study selection as suggested by the reviewer (page 4 para 6).
Inclusion criteria for study selection		
14	1. Line 21: You state that you will include unpublished PD trials. However, you do not include grey literature. Could you please clarify how you will search for unpublished PD trials? 2. Line 22: “unpublished for a maximum of 5 years following study completion” – This is unclear, I suggest explaining this and giving a rationale. 3. Line 23: Why are phase 1 studies and conference abstracts excluded? Could you provide a justification for this?	We have clarified the source of unpublished trials in the study selection section and provided rationales for exclusion criteria (page 4 last paragraph): “We have used the Population, Intervention, Comparator, Outcome, and Study design (PICOS) framework to develop study eligibility criteria aiding in the identification of PD trials (Table 1). Records in English, including published and planned as well as unpublished studies identified within clinicaltrials.gov will be fully extracted. Phase 1 studies will be excluded as the focus of the review is to inform the design of a phase 2/3 study seeking to evidence efficacy rather than safety/tolerability which is the focus of phase 1 studies. Studies for which only conference abstracts are available will be excluded as information within abstracts is too limited for data

		extraction.”
15	4. Table 1 population: I suggest shortening the inclusion criterion to “participants with idiopathic PD”	We have adjusted this as requested (page 5).
16	5. Table 1 intervention: Please provide a rationale for excluding DBS	Thank you for this comment. Our preliminary searches identified deep brain stimulation as a frequent non-relevant hit. Whilst it is possible that the introduction of this exclusion may introduce a small chance of missing DBS related disease modification studies, this decision was mainly based on practicality. We have added this to the study limitations (page 3). All other interventions will be judged on authors’ intent. We have not made any further changes to the manuscript as this is outlined in table 2 which describes the rationale for electronic search keywords (page 7).
17	6. Table 1 comparator, outcome, study design: It would be helpful to provide a definition for phase 2 and 3 trials as these are used differently in the literature. Regarding the outcomes: please also define efficacy and safety outcomes and provide a rationale for excluding pure safety trials as phase 2 trials are safety trials by nature.	Many thanks for this comment. We now include a rationale for excluding pure safety trials and define our screening criteria related to trial phases (page 5): “Outcome - The focus of the review is on phase 2 and 3 efficacy trials and therefore trials will have to include at least one efficacy outcome. Pure safety trials will be excluded. Study design - Only phase 2 and 3 trials will be included as this work will be carried out to support the design of a phase 2/3 platform trial. For article screening purposes, trial phases as stated by article or registry author will be used.”
	Search Methods for Identification of Studies	
18	1. Line 17: please update the date and search as this is a protocol this should be dated after publishing the protocol	This was updated to 1st of October 2023
19	2. Lines 20-25: This is a protocol; therefore, the search should not have been conducted yet. Furthermore, please use future tense for what you are planning to do	We apologise for the misunderstanding. This paragraph refers to our search development strategy and not the conduct of the searches themselves. We have clarified this through an additional section heading and by re-structuring this paragraph. We have not amended the tense throughout the paragraph, as the search development has already

		been conducted (although the searches themselves have not been completed). We hope that this is now clearer.
20	3. Lines 24-26: I suggest writing “inclusion criteria” instead of “PICOS criteria”	This was amended to “eligibility criteria”
21	4. Line 29: You write that only studies with a status marked as “completed” will be included. Does this mean that you will exclude studies where the status was possibly not updated to “completed” even though the study has been completed and possibly the results are already published? It would be helpful if you could clarify this.	Thank you for this comment. Yes, only studies with a “completed” status were included as this subset of studies has the highest likelihood of being published. We have clarified this in the manuscript. The aim of this validation strategy was to generate a random sample of eligible studies which could be used to determine search term efficiency. We would argue that missing a small percentage of studies at random due to a delay in study updates on the registry will have negligible impact on the described validation procedure. We have attempted to restructure the description of our search validation method to make it easier to follow (page 6)
22	5. Line 38: Is this validation process based on the PRESS (peer review of electronic search strategies) guidelines? If so, I would suggest referencing these guidelines and explaining in more detail how you will implement this.	Thank you for pointing us towards these very helpful guidelines, which we were not aware of when developing the search strategy. We will ensure the search is peer reviewed prior to commencement of searches and have added the following sentence to the manuscript (page 6 final para): “Search strategies will be peer reviewed following PRESS guidelines”
	Data extraction and management	
23	1. Table 3 outcome measures: It is unclear what you mean by “outcome domain”. Do you have a reference for this? E.g. Dodd et al. 2018? It would be helpful to clarify this.	The following sentence was added to the Data collection and planned analysis section under outcome measures (page 10 para 1): “ Outcome domains will be defined using the National Institute for Neurological Disorders and Stroke Common Data Elements (NINDS-CDE) for Parkinson’s domains and sub-domains (27)”
24	2. Table 3 outcome measures: Are you only looking at outcome measures or at outcomes or at both? It might be that you mean outcomes instead of outcome measures?	We will be extracting outcome measures listed by studies and classify these into outcome domains for analysis. We have clarified the terminology within Table 3 (page 8), Outcome measures and outcome measures success sections within the manuscript.

25	3. Table 3 study design: It could be interesting to look at the frequency of measurement in addition to study duration; so how many follow-ups did they conduct and at what timepoints.	Thank you for this recommendation. We have added these as additional exploratory extraction fields as follows (page 8): We have added number and frequency of follow-ups within the trial design extraction domain, as this will be the number of total follow-ups within the study (i.e. all assessments). We have also added this into our analysis plan for Study size, duration and withdrawals as follows (page 10): “We will perform a descriptive analysis to summarize study size, duration, number and frequency of follow-ups and withdrawals.”
Risk of bias assessment		
26	1. The Jadad scale seems to be oversimplistic as there is too much emphasis on blinding. I suggest choosing another tool or provide a strong rationale for using the Jadad scale. 2. Line 56: Why are you planning on analysing only selected studies for risk of bias? Please provide a strong rationale for this and explain how you will select these.	We have changed the proposed review to a scoping review. Therefore an assessment of risk of bias is no longer required. The risk of bias assessment section has been removed (page 8-9)
Data collection and planned analysis		
27	1. Lines 15-19: This is another aim / objective. I suggest moving this to the introduction section and align it with the objective in the abstract.	We have aligned these throughout abstract and introduction. Abstract (Page 2, para 1): “It is not clear whether consensus is emerging regarding trial design choices. Here we report the protocol of a scoping review that will provide a contemporary update on trial design variability for disease-modifying interventions in PD.” Page 2 Para 3: “This work will provide an overview of variation and emerging trends in trial design choices for disease modifying trials of Parkinson’s.” Introduction page 4, para 4: “Here, we report on our protocol to systematically chart the design of phase 2 and 3 disease modifying trials in PD with the view of informing the design of a randomised, controlled phase 3 adaptive multi-arm multi-stage platform trial for disease-modifying therapies in PD. The review will provide an overview of trial design characteristics such as participant selection,

		stratification/minimisation criteria, trial size, duration and outcome measures to assess whether there are emerging trends on trial design choices.” Extracting and charting results (page 9): “The aim of this review is to map emerging trends in trial design choices such as participant selection, stratification/minimisation criteria, trial size, duration and outcome measures.”
28	2. Eligibility criteria, lines 45-47: Please provide the reference for this scoping review	The reference to this work is now included (page 9, last para)
29	3. Outcome measures lines 54-55: In table 3 you differentiate between primary and secondary outcomes; therefore, this is contradictory. I suggest providing a clarification or amending this section accordingly.	We amended the table to Primary outcome measures and other outcome measures (page 8 and page 10).
30	4. Outcome measures line 59: Please explain how you will analyse this. Additionally, in the introduction you stated that there might be more suitable, validated scales other than the UPDRS. What is your rationale for only analysing the two listed scales?	We have clarified the proposed descriptive analysis. We will extract and provide data for all outcome measures used in all trials but will summarise these within the report using outcome domains as described in the manuscript (page 10 para 1): “We will provide full data on frequency of all outcome measures and will summarise these as follows: the frequency of outcome domains used as primary outcome measures in phase 2 and 3 trials. Outcome domains will be defined using the National Institute for Neurological Disorders and Stroke Common Data Elements (NINDS-CDE) for Parkinson’s domains and sub-domains (26). We will additionally chart the use of the most common outcome measure scale, the Unified Parkinson’s Disease Rating Scale (UPDRS) and the Movement Disorder Society version (MDS-UPDRS), reporting on the use of its parts and part combinations as primary. This is particularly important in the light of a recent report by the scale author’ affirming a recommendation against the combination of part 3 with other parts of the scale (27).”
	Development of consensus	
31	It is unclear how the stated description relates to the development of consensus. I suggest amending this paragraph.	We have changed our terminology from consensus to “Study design trends over time” (page 10, para 4).
	Assessment of Reporting Bias	

32	Please provide a rationale for excluding these trial registry entries. I recommend considering that if you do exclude them, this could add to a risk of reporting bias in your review.	We have amended this methodology and will now include studies that remained unpublished but will report on the number of studies that have remained unpublished for over 5 years. "We will report on the number of studies that have been completed for longer than five years without a published peer-reviewed results report to provide an indication of potential reporting bias." (page 10, para 5)
	Patient and Public Involvement	
33	1. Line 37: I assume that you mean two instead of three patients? Please adapt accordingly.	Yes. We have adapted this (page 10 final para)
	Funding statement	
34	According to the PRISMA-P checklist #5c please describe the role of funder(s), sponsor(s) and/or institution(s), if any, in developing the protocol.	We have added following to the funding statement: "The funder was not actively involved in the development of this protocol." (page 11, final para)

VERSION 2 – REVIEW

REVIEWER	Hirschwald, Julia Trinity College Dublin, Clinical Speech and Language Studies
REVIEW RETURNED	11-Oct-2023

GENERAL COMMENTS	Many thanks for the thorough revision of the manuscript. Please find below some minor last comments for further improvement of the study protocol. Abstract  - As you are now planning a scoping review instead of a systematic review, I would recommend using the PCC instead of PICOS format as this is suggested to use for scoping reviews (for example see the JBI manual https://jbi-global-wiki.refined.site/space/MANUAL/4687342/Chapter+11%3A+Scoping+reviews) - "Two independent reviewers will screen study titles and abstracts for eligibility, with disagreements being resolved through discussion or by a third reviewer where necessary." – The step of full text screening is missing here. I assume you are planning on doing this with two independent reviewers as well? If so, I would recommend adding this here. - "This work will provide an overview of variation and emerging trends in trial design choices for disease modifying trials of Parkinson's." – I would recommend using the abbreviation PD instead of Parkinson's here - "We will report our findings using the Preferred Reporting Items for Systematic Reviews and Meta-Analyses (PRISMA) Scoping review extension." – I would recommend stating this sentence in the methods section of the abstract. The correct abbreviation would be PRISMA-ScR.
--

	Introduction  - Page 3, line 30: I would recommend to double check that all “Parkinson’s” or “Parkinson’s disease” are replaced by PD from this point onwards; line 56 same for DMT Methods and analysis  - Page 4, line 17: I would suggest writing that the protocol is written in accordance with the PRISMA-ScR guideline - Page 4, line 18: see comment on abstract regarding PCC format or potentially provide a rationale for why using PICOS format - Tables: please provide explanation for abbreviations used in tables where applicable below each table - Page 9, line 5: I would recommend taking out the aim here, as this should have been sufficiently explained at the end of the introduction already
--	--

VERSION 2 – AUTHOR RESPONSE

Reviewer 2 Comments	Response
Abstract	
As you are now planning a scoping review instead of a systematic review, I would recommend using the PCC instead of PICOS format as this is suggested to use for scoping reviews (for example see the JBI manual https://jbi-global-wiki.refined.site/space/MANUAL/4687342/Chapter+11%3A+Scoping+reviews)	Many thanks for this suggestion. However, as our research question relates to trials of clinical interventions we feel the PICOS is a more appropriate tool to define scope of this review.
“Two independent reviewers will screen study titles and abstracts for eligibility, with disagreements being resolved through discussion or by a third reviewer where necessary.” – The step of full text screening is missing here. I assume you are planning on doing this with two independent reviewers as well? If so, I would recommend adding this here.	We have changed this to “Two independent reviewers will screen study titles, abstracts and full text for eligibility, with disagreements being resolved through discussion or by a third reviewer where necessary. “ (Abstract, paragraph 2)
This work will provide an overview of variation and emerging trends in trial design choices for disease modifying trials of Parkinson’s.” – I would recommend using the abbreviation PD instead of Parkinson’s here	We have changed this to PD. (abstract, paragraph 3)

“We will report our findings using the Preferred Reporting Items for Systematic Reviews and Meta-Analyses (PRISMA) Scoping review extension.” – I would recommend stating this sentence in the methods section of the abstract. The correct abbreviation would be PRISMA-ScR.	The text has been amended as suggested. (Abstract, paragraph 2 and 3)
Introduction	
Page 3, line 30: I would recommend to double check that all “Parkinson’s” or “Parkinson’s disease” are replaced by PD from this point onwards; line 56 same for DMT	We have changed all references to “Parkinson’s” or “Parkinson’s disease” to “PD” where appropriate and ensured the term DMT was used for disease modifying therapies throughout.
Method and Analysis	
Page 4, line 17: I would suggest writing that the protocol is written in accordance with the PRISMA-ScR guideline	Page 4, line 17: Thank you for this suggestion. We have changed the sentence to “The scoping review protocol presented here was written in accordance with PRISMA scoping review (PRISMA-ScR) guidelines”
Page 4, line 18: see comment on abstract regarding PCC format or potentially provide a rationale for why using PICOS format	Please see response to comment 1.
Tables: please provide explanation for abbreviations used in tables where applicable below each table	We have checked the tables for abbreviations and provided explanations or removed abbreviations as appropriate. See page 4,5 and 8
Page 9, line 5: I would recommend taking out the aim here, as this should have been sufficiently explained at the end of the introduction already	Page 9, Line 5: Text has been removed as suggested;the comment on separating reporting by trial phases has been moved into the “Study phase” section